

# A mathematical formulation and an NSGA-II algorithm for minimizing the makespan and energy cost under time-of-use electricity price in an unrelated parallel machine scheduling

Marcelo F. Rego[1,2], Júlio Cesar E. M. Pinto[2], Luciano P. Cota[3] and Marcone J. F. Souza[2]

[1] Department of Computing, Universidade Federal dos Vales dos Jequitinhonha e Mucuri, Diamantina, Minas Gerais, Brazil
[2] Department of Computing, Universidade Federal de Ouro Preto, Ouro Preto, Minas Gerais, Brazil
[3] Instituto Tecnológico Vale, Ouro Preto, Minas Gerais, Brazil

## ABSTRACT

In many countries, there is an energy pricing policy that varies according to the time-of-use. In this context, it is financially advantageous for the industries to plan their production considering this policy. This article introduces a new bi-objective unrelated parallel machine scheduling problem with sequence-dependent setup times, in which the objectives are to minimize the makespan and the total energy cost. We propose a mixed-integer linear programming formulation based on the weighted sum method to obtain the Pareto front. We also developed an NSGA-II method to address large instances of the problem since the formulation cannot solve it in an acceptable computational time for decision-making. The results showed that the proposed NSGA-II is able to find a good approximation for the Pareto front when compared with the weighted sum method in small instances. In a large number of instances, NSGA-II outperforms, with a 95% confidence level, the MOGA and NSGA-I multi-objective techniques concerning the hypervolume and hierarchical cluster counting metrics. Thus, the proposed algorithm finds non-dominated solutions with good convergence, diversity, uniformity, and amplitude.

# INTRODUCTION

The industrial sector is one of the largest consumers of energy in the world. According to *EIA (2016)*, this sector consumes around 54% of the total energy distributed globally.

Among the various forms of energy used by the manufacturing industry, electricity has been one of the most consumed. In China, for example, this sector consumes about 50% of the electricity produced in the country (*Liu et al., 2014*).

In recent years, electricity prices have continuously increased for manufacturing companies in industrialized countries (*Willeke, Ullmann & Nyhuis, 2016*). In Norway, the

Corresponding authors
Marcelo F. Rego,
marcelofr@ufvjm.edu.br
Luciano P. Cota,
luciano.p.cota@itv.org

industrial electricity price, including taxes, increased by 47% between 2017 and 2018 (*BEIS, 2020*). This increase has an impact on production costs and can reduce the competitiveness of companies. In countries that implement a pricing policy so that the energy price depends on the time-of-use, the reduction of electricity costs can occur through production planning that prioritizes periods when energy is less expensive.

However, few studies address scheduling problems in which the energy price depends on the time-of-use tariffs. Among them, we mention *Ebrahimi et al. (2020)*, *Zeng, Che & Wu (2018)*, *Wang et al. (2016)*, *Shrouf et al. (2014)*, *Zhang et al. (2014)*, where the objective includes to minimize the total energy cost.

On the other hand, among several scheduling environments, the unrelated parallel machine one has received much attention recently, given its wide applicability in the industry (*Cota et al., 2019*). In terms of performance measures, makespan minimization is one of the most common because this criterion aims at the good utilization of the machines (*Pinedo, 2016*). Lastly, the sequence-dependent setup times appear in many industrial and service applications (*Kopanos, Lanez & Puigjaner, 2009*). However, as far as we know, no work reported in the literature addresses the unrelated parallel machine scheduling problem with sequence-dependent setup times (UPMSP-SDS), considering minimizing the makespan and the total energy cost. This paper, therefore, aims to fill this gap.

The main contributions of this work are the following: (i) introducing a new bi-objective unrelated parallel machine scheduling problem; (ii) introducing a new mixed-integer linear programming formulation able to solve small-scale instances of this problem; (iii) proposing an adapted version of the NSGA-II algorithm to treat large-scale instances of this problem; (iv) creating a set of instances for this problem; (v) performing an experimental study of the proposed methods.

We organized the remainder of this article as follows: In "Literature Review", we review the literature. In "Problem Statement", we detail the problem addressed. In "Weighted Sum Method", we introduce the proposed mathematical model. In "Proposed NSGA-II", we show the adaptation of the NSGA-II algorithm to the problem. In "Computational Experiments", we report the computational results, which include a comparison of the results of the proposed algorithm with the exact method on small instances and a comparison with other multi-objective algorithms on large instances. Finally, we present the conclusions and directions for future work in "Conclusions".

## LITERATURE REVIEW

Here, we present a literature review with previous research that addressed scheduling problems and considered objectives related to this work.

Some studies address the scheduling problem only to minimize energy consumption. For example, *Shrouf et al. (2014)* proposed a mathematical model for the scheduling problem on a single machine. However, the model cannot solve large instances within a reasonable computational time for decision-making. For this reason, they also proposed a genetic algorithm. The computational results indicated the possibility of reducing energy consumption by up to 30% when they compared the genetic algorithm solution and the "as soon as possible" heuristic solution. *Tsao, Thanh & Hwang (2020)* presented a fuzzy

model integrated into a genetic algorithm for a problem similar to that described previously. They tested their method in instances of up to 200 jobs. The results indicated a 4.20% reduction in total energy consumption compared to the traditional genetic algorithm.

Other studies address the scheduling problem and consider a second objective beyond minimizing energy consumption. *Cota et al. (2018)* proposed a mathematical model and applied a mathematical heuristic called multi-objective smart pool search for the UPMSP-SDS. The objective functions are to minimize the makespan and the total energy consumption. In the experiments, they used a set of instances with up to fifteen jobs, and five machines randomly generated. They adopted hypervolume and set coverage metrics to compare the proposed algorithm with the ε-constraint exact method. They showed that the objectives are conflicting and that energy consumption strongly influences the solution's quality. *Cota et al. (2019)* introduced the MO-ALNS and MO-ALNS/D algorithms to handle instances of up to 250 jobs and 30 machines of the same problem described previously. The MO-ALNS algorithm is a multi-objective version of the Adaptive Large Neighborhood Search–ALNS (*Ropke & Pisinger, 2006*), and MO-ALNS/D combines the multi-objective MOEA/D (*Zhang & Li, 2007*) with ALNS. The MO-ALNS/D algorithm was able to find better results than MO-ALNS in most instances in the hypervolume, set coverage, and Hierarchical Cluster Counting (HCC) (*Guimaraes, Wanner & Takahashi, 2009*) metrics. (*Wu & Che, 2019*) proposed a memetic differential evolution (MDE) algorithm for the UPMSP in which the objectives are also to minimize the makespan and the total energy consumption. The computational results showed that the proposed approach provides a significant improvement over the basic DE. Also, the MDE outperforms the SPEA-II and NSGA-II algorithms. (*Liang et al., 2015*) presented the Ant Colony Optimization algorithm with the Apparent Tardiness Cost (ACO-ATC) rule for the UPMSP seeking to minimize the total tardiness and the energy consumption. In this problem, machines need to wait until jobs are ready. However, it is necessary to decide whether the machine remains on or off during the wait. Turning off the machine to wait for the job to be ready saves energy. On the other hand, keep on the machine while waiting for the job saves time because it eliminates the need to prepare the machine again. They compared the ACO-ATC results with the classic ACO and a GRASP-based algorithm (*Feo & Resende, 1995*). The proposed algorithm was better than the other approaches in most of the tested instances.

There are studies that only address the minimization of the total energy cost. *Ding et al. (2016)* presented two approaches to UPMSP: the first introduces a time-interval-based Mixed Integer Linear Programming (MILP) formulation. The second consists of a reformulation of the problem using the Dantzig-Wolfe decomposition and a column generation heuristic. According to the results, the MILP formulation overcame the column generation method in terms of solution quality and execution time when electricity prices stay stable for a relatively long period. On the other hand, the column generation method performed better when the electricity price frequently changed (*i.e.*, every half hour). *Cheng, Chu & Zhou (2018)* improved the formulation by *Ding et al. (2016)* and performed computational experiments with 120 randomly generated instances to compare

**Table 1 Summary of characteristics addressed by our work compared to literature studies.**

| Reference | Unrelated parallel machines | Sequence-dependent setup | Makespan | Total energy cost | Time-of-use | Multi-objective | Exact method | Metaheuristic method |
|---|---|---|---|---|---|---|---|---|
| Shrouf et al. (2014) | | | | √ | | | √ | √ |
| Liang et al. (2015) | √ | | | | | √ | √ | √ |
| Ding et al. (2016) | √ | | | √ | √ | | √ | |
| Kurniawan et al. (2017) | √ | | √ | √ | √ | √ | | √ |
| Cota et al. (2018) | √ | √ | √ | | | √ | √ | |
| Cheng, Chu & Zhou (2018) | √ | | | √ | √ | | √ | |
| Zeng, Che & Wu (2018) | | | | √ | √ | √ | √ | √ |
| Cota et al. (2019) | √ | √ | √ | | | √ | | √ |
| Wu & Che (2019) | √ | | √ | | | √ | | √ |
| Cheng, Wu & Chu (2019) | √ | | √ | √ | √ | √ | √ | |
| Tsao, Thanh & Hwang (2020) | | | | √ | | | √ | √ |
| Saberi-Aliabad, Reisi-Nafchi & Moslehi (2020) | √ | | | √ | √ | | √ | √ |
| Our proposal | √ | √ | √ | √ | √ | √ | √ | √ |

the two formulations. The results showed that the new formulation achieves better results concerning the solution quality and execution time. *Saberi-Aliabad, Reisi-Nafchi & Moslehi (2020)* proposed the fix-and-relax heuristic algorithm in two stages for this same problem. In the first stage, jobs are assigned to the machines, and the second one solves a scheduling problem on simple machines. They tested its method in 20 instances randomly generated following the same parameter values as other previous studies. They compared the proposed method with the algorithms of *Che, Zhang & Wu (2017)* and *Cheng, Chu & Zhou (2018)*. The results showed that the fix-and-relax algorithm overcame the others.

Finally, we present studies that address the scheduling problem considering minimizing the energy cost combined with another objective. *Zeng, Che & Wu (2018)* dealt with the bi-objective uniform parallel machine scheduling to minimize the total energy cost and the number of machines. They proposed a new mathematical model and a heuristic algorithm for it. The computational results showed that the heuristic method generates high-quality solutions in a reasonable time limit for instances of up to 5,000 jobs. *Cheng, Wu & Chu (2019)* presented a mathematical formulation and a genetic algorithm for the UPMSP. They considered the objective of minimizing the weighted sum of makespan and total electricity cost. The results presented by their formulation overcome that of the genetic algorithm in terms of solution quality. *Kurniawan et al. (2017)* proposed a genetic algorithm with a delay mechanism for the UPMSP to minimize the weighted sum of makespan and total energy cost. The proposed algorithm handled instances of up to 30 jobs and 15 machines. The results showed that the proposed method provided better solutions than the classical genetic algorithm.

Although there are studies correlated to ours, of our knowledge, there is no work addressing the unrelated parallel machine scheduling problem with sequence-dependent

**Table 2 Decision and auxiliary variables for the problem.**

| Name | Description |
|---|---|
| $X_{ijhl}$ | Binary variable that assumes value 1 if the job $j$ is allocated on the machine $i$ at time $h$ and in the operation mode $l$, and value 0, otherwise |
| $PEC_t^{on}$ | Partial Energy Cost (\$) during the on-peak in day $t \in D$ |
| $PEC_t^{off}$ | Partial Energy Cost (\$) during the off-peak in day $t \in D$ |
| $C_{max}$ | The maximum completion time of the jobs, also known as makespan |
| $TEC$ | Total Energy Cost (\$) |

setup times to minimize the makespan and the total energy cost. Table 1 summarizes the characteristics of scheduling problems treated by our work compared to literature references.

# PROBLEM STATEMENT

To define the UPMSP-SDS, we characterize the problem in this section and introduce a MILP formulation to solve it.

The following are the characteristics of the problem addressed in this work:
- There are a set $N = \{1, \ldots, n\}$ of jobs, a set $M = \{1, \ldots, m\}$ of machines, and a set $L = \{1, \ldots, o\}$ of different operation modes, such that each operation mode $l \in L$ is associated with a multiplication factor of speed $v_l$ and a multiplication factor of power $\lambda_l$;
- The machines are unrelated parallel. In other words, the processing time of job $j \in N$ can be different on each machine $i \in M$;
- There is a planning horizon that consists of a set of $H = \{0, \ldots, |H|\}$ of time instants, and we must perform all jobs within this horizon;
- All jobs are available to be processed at the beginning of the planning horizon $h = 0$;
- Each job $j \in N$ must be allocated to exactly one machine $i \in M$;
- There is a processing time $p_{ij}$ to process a job $j \in N$ on a machine $i \in M$;
- There is a sequence-dependent setup time $S_{ijk}$ to execute a job $k \in N$ after another job $j \in N$ on a machine $i \in M$;
- Each machine $i \in M$ has a power $\pi_i$ at normal operating speed;
- The operation mode $l \in L$ of each job determines the multiplication factor of power ($\lambda_l$). It also determines the multiplication factor of speed ($v_l$), which, in turn, is related to the execution time of each job;
- There is a set $D$ of days on the planning horizon $H$;
- Each day is discretized into $sizeD$ time intervals. For example, for discretizing a day in minutes, $sizeD = 1,440$; for the discretization of one day in hours, $sizeD = 24$;
- To each day $t \in H$, we have a peak hour, which starts at the time $startp_t \in H$ and ends at the time $endp_t \in H$;
- $ET^{off}$ and $ET^{on}$ represent the energy tariff (\$/KWh) in off-peak hours and on-peak hours, respectively.

Table 2 presents the decision and auxiliary variables that are needed to model the problem.

Thus, we can define the problem through Eqs. (1)–(12).

$$\min C_{\max} \tag{1}$$

$$\min TEC \tag{2}$$

Subject to:

$$\sum_{i=1}^{m} \sum_{l=1}^{o} \sum_{h=0}^{|H|-\left\lceil \frac{P_{ij}}{V_l} \right\rceil} X_{ijhl} = 1 \quad \forall j \in N \tag{3}$$

$$X_{ijhl} + \sum_{u=h}^{\min\left(h+\left\lceil \frac{P_{ij}}{V_l} \right\rceil + S_{ijk}-1,|H|\right)} \sum_{l_1=1}^{o} X_{ikul_1} \leq 1 \quad \forall i \in M, j \in N, k \in N, l \in L, j \neq k \tag{4}$$

$$C_{\max} \geq X_{ijhl} \times \left[ h + \left\lceil \frac{P_{ij}}{V_l} \right\rceil \right], \quad \forall i \in M, j \in N, h \in H, l \in L \tag{5}$$

$$PEC_t^{on} \geq \sum_{i=1}^{m} \sum_{j=1}^{n} \sum_{l=1}^{o} \frac{\lambda_l \times \pi_i \times ET^{on} \times 24}{sizeD} \times \tag{6}$$

$$\left\{ \sum_{h=sizeD \times (t-1)}^{startp_t - 1} X_{ijhl} \times \left[ \max\left( 0, \min\left( h + \left\lceil \frac{P_{ij}}{V_l} \right\rceil - 1, endp_t \right) - (startp_t - 1) \right) \right] \right.$$

$$\left. + \sum_{h=startp_t}^{endp_t - 1} X_{ijhl} \times \left[ \min\left( h + \left\lceil \frac{P_{ij}}{V_l} \right\rceil, endp_t + 1 \right) - h \right] \right\} \quad \forall t \in D$$

$$PEC_t^{off} \geq \sum_{i=1}^{m} \sum_{j=1}^{n} \sum_{l=1}^{o} \frac{\lambda_l \times \pi_i \times ET^{off} \times 24}{sizeD} \times \tag{7}$$

$$\left\{ \sum_{h=sizeD \times (t-1)}^{startp_t - 1} X_{ijhl} \times \left[ \min\left( h + \left\lceil \frac{P_{ij}}{V_l} \right\rceil, startp_t \right) - h + \max\left( 0, h + \left\lceil \frac{P_{ij}}{V_l} \right\rceil - endp_t - 1 \right) \right] \right.$$

$$+ \sum_{h=startp_t}^{endp_t - 1} X_{ijhl} \times \left[ \max\left( 0, h + \left\lceil \frac{P_{ij}}{V_l} \right\rceil - endp_t - 1 \right) \right]$$

$$\left. + \sum_{h=endp_t}^{|H|-1} X_{ijhl} \times \left\lceil \frac{P_{ij}}{V_l} \right\rceil \right\} \quad \forall t \in D$$

$$TEC \geq \sum_{t=1}^{sizeD} \left( PEC_t^{off} + PEC_t^{on} \right) \tag{8}$$

$$X_{ijhl} \in \{0, 1\} \quad \forall i \in M, j \in N, h \in H, l \in L \tag{9}$$

$$C_{\max} \geq 0 \tag{10}$$

$$PEC_t^{off} \geq 0 \quad \forall t \in D \tag{11}$$

$$PEC_t^{on} \geq 0 \quad \forall t \in D \tag{12}$$

The objectives of the problem are to minimize, simultaneously, the makespan and the total energy cost, defined by Eqs. (1) and (2), respectively. The set of constraints (3) ensures that every job $j \in J$ is allocated on a machine has a single operation mode, and ends its execution inside the planning horizon. Constraints (4) define that if the job $k$ is assigned to

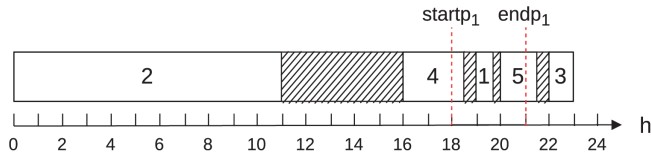

**Figure 1  Example to illustrate the calculation of the energy cost on a machine.**

the machine $i$ immediately after the job $j$, then the start time of the job $k$ must be greater than the sum of the end time of the job $j$ and the setup time between them. It is important to highlight that for the previous model to be valid, the setup and processing times must satisfy the triangular inequality, as defined by *Rosa & Souza (2009)*. The set of constraints (5) determines a lower bound for the makespan. Constraints (6) and (7) define a lower bound for the partial energy cost in the on-peak hours ($PEC_{on}$) and the total energy cost for off-peak hours ($PEC_{off}$), respectively. It is important to note that a job can be partially performed in the on-peak hours and partially in the off-peak hours and that the total energy cost is directly related to the energy price and the job execution time. Constraint (8) ensures a lower bound for the total energy cost. Constraints (9)–(12) define the domain of the decision and auxiliary variables of the problem.

The calculation of the energy cost of a job $j$ depends on its execution time during the on-peak and off-peak time. Thus, there are six possible cases:

Case 1: The job $j$ starts and ends before the on-peak hours;

Case 2: The job $j$ starts before the on-peak hours and ends in the on-peak hours;

Case 3: The job $j$ starts and ends in the on-peak hours;

Case 4: The job $j$ starts during the on-peak hours and ends after the on-peak hours;

Case 5: The job $j$ starts and ends after the on-peak hours;

Case 6: The job $j$ starts before the on-peak hours and ends after the on-peak hours.

To illustrate cases 1 to 5, let Fig. 1. It shows the execution of five jobs $N = \{2, 4, 1, 5, 3\}$ in the scheduling of a single machine $i = 1$ in a single operation mode $l = 1$ on day $t = 1$ of the planning horizon. Let the start of the on-peak hours ($startp_1$) equal to 18; the end of the on-peak hours ($endp_1$) equal to 21; the multiplication factor of power ($\lambda_l$) equal to 1; the energy consumption of machine at normal operation ($\pi_1$) equal to 100; the energy tariff in the on-peak hours ($ET^{on}$) equal to $0.10\$/KWh$ and in the off-peak hours ($ET^{off}$) equal to $0.05\$/KWh$; the multiplication factor of speed $v_l$ equal to 1. In this example, we consider discretization in hours. This figure shows that jobs 4, 1, and 5 are performed in the on-peak hours, partially or totally, and jobs 2 and 3, in turn, in the off-peak hours.

For this example, Eqs. (6) and (7) are reduced to Eqs. (13) and (14) below:

$$PEC_1^{on} = \sum_{j=1}^{n} \underbrace{\frac{1 \times 100 \times 0.10 \times 24}{24}}_{Parcel\ 1(a)} \times \tag{13}$$

**Table 3 Energy cost by job in the example of Fig. 1.**

| # Job | Case | Contr. on-peak hours | Contr. off-peak hours |
|-------|------|----------------------|-----------------------|
| 2 | 1 | $\underbrace{10}_{(a)} \times (\underbrace{0}_{(b)} + \underbrace{0}_{(c)}) = 0$ | $\underbrace{5}_{(d)} \times (\underbrace{11}_{(e)} + \underbrace{0}_{(f)} + \underbrace{0}_{(g)}) = 55$ |
| 4 | 2 | $\underbrace{10}_{(a)} \times (\underbrace{1}_{(b)} + \underbrace{0}_{(c)}) = 10$ | $\underbrace{5}_{(d)} \times (\underbrace{2}_{(e)} + \underbrace{0}_{(f)} + \underbrace{0}_{(g)}) = 10$ |
| 1 | 3 | $\underbrace{10}_{(a)} \times (\underbrace{0}_{(b)} + \underbrace{1}_{(c)}) = 10$ | $\underbrace{5}_{(d)} \times (\underbrace{0}_{(e)} + \underbrace{0}_{(f)} + \underbrace{0}_{(g)}) = 0$ |
| 5 | 4 | $\underbrace{10}_{(a)} \times (\underbrace{0}_{(b)} + \underbrace{1}_{(c)}) = 10$ | $\underbrace{5}_{(d)} \times (\underbrace{0}_{(e)} + \underbrace{1}_{(f)} + \underbrace{0}_{(g)}) = 5$ |
| 3 | 5 | $\underbrace{10}_{(a)} \times (\underbrace{0}_{(b)} + \underbrace{0}_{(c)}) =$ | $\underbrace{5}_{(d)} \times (\underbrace{0}_{(e)} + \underbrace{0}_{(f)} + \underbrace{1}_{(g)}) = 5$ |

$$PEC_1^{off} = \sum_{j=1}^{n} \underbrace{\frac{1 \times 100 \times 5 \times 24}{24}}_{Parcel\ 1(d)} \times \left\{ \underbrace{\left[ \sum_{h=0}^{18-1} X_{1jh1} \times \left[ \max\left(0, \min\left(h + \left\lceil \frac{P_{1j}}{1} \right\rceil - 1, 21\right) - (18-1)\right) \right] \right]}_{Parcel\ 2(b)} + \underbrace{\sum_{h=18}^{21-1} X_{1jh1} \times \left[ \min\left(h + \left\lceil \frac{P_{1j}}{1} \right\rceil, 21+1\right) - h \right]}_{Parcel\ 3(c)} \right\} \times \left\{ \underbrace{\left[ \sum_{h=0}^{18-1} X_{1jh1} \times \left[ \min\left(h + \left\lceil \frac{P_{1j}}{1} \right\rceil, 18\right) - h + \max\left(0, h + \left\lceil \frac{P_{1j}}{1} \right\rceil - 21 - 1\right) \right] \right]}_{Parcel\ 2(e)} + \underbrace{\sum_{h=18}^{21-1} X_{1jh1} \times \left[ \max\left(0, h + \left\lceil \frac{P_{1j}}{1} \right\rceil - 21 - 1\right) \right]}_{Parcel\ 3(f)} + \underbrace{\sum_{h=21}^{24-1} X_{1jh1} \times \left\lceil \frac{P_{1j}}{1} \right\rceil}_{Parcel\ 4(g)} \right\}$$  (14)

Table 3 illustrates the contribution of each job to the total energy cost, according to the example in Fig. 1. The column "# Job" represents the job, the column "Case" shows the contemplated case, and the columns "Contr. on-peak" and "Contr. off-peak" show the contributions of the job to the energy cost of each job in the on-peak and off-peak hours, respectively.

The total energy cost found to the schedule shown in Fig. 1 is 105.

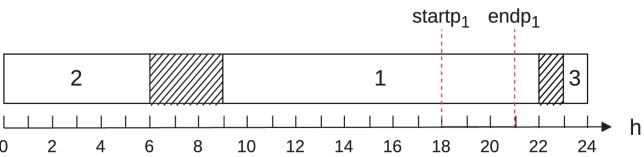

**Figure 2 Schedule example for case 6.**

To illustrate case 6, consider Fig. 2. It shows the execution of three jobs $N = \{2, 1, 3\}$ on a single machine $i = 1$ in operating mode $l = 1$ during day $t = 1$ of the planning horizon. Let also the start of the on-peak hours ($startp_1$) equal to 18; the end of the on-peak hours ($endp_1$) equal to 21; the multiplication factor of power ($\lambda_l$) equal to 1; the energy consumption of machines at normal operation ($\pi_1$) equal to 100; the energy tariff in the on-peak hours ($ET^{on}$) equal to 0.10 $/KWh$ and in the off-peak hours ($ET^{off}$) equal to 0.05 $/KWh$; and the multiplication factor of speed equal to 1. Such as in the previous example, we consider discretization in hours. This figure shows that job 1 is performed in the on-peak hours and jobs 2 and 3, in turn, in the off-peak hours.

The contribution of the job 1 to the energy cost in the on-peak hours is 30, and the contribution to the cost in the off-peak hours is 50.

Thus, calculating similarly to the previous example, we conclude that the total energy cost for the schedule shown in Fig. 2 is 155.

# WEIGHTED SUM METHOD

We used the weighted sum method (*Marler & Arora, 2004*) to solve the multi-objective optimization problem addressed using a mathematical programming solver. This method converts the multi-objective problem into a single objective problem using the weighted sum of the objectives.

For this, consider Eq. (15):

$$\min \quad z(X) = \left[ \alpha \times \left( \frac{C_{\max}}{|H|} \right) + (1 - \alpha) \times \left( \frac{TEC}{Cost_{\max}} \right) \right] \tag{15}$$

where:
- $\alpha$: real number in range [0, 1];
- $|H|$: represents the cardinality of the set $H$;
- $Cost_{\max}$: is the estimate for the maximum energy cost used to normalize the total energy cost. It is calculated using a heuristic, as shown in "Initial Population";

The problem constraints are those defined by Eqs. (3)–(12).

Algorithm 1 describes all the steps of the weighted sum method implemented.

Algorithm 1 receives as input: the set $\Delta$ with the values for $\alpha$ and the time limit. In line, we initialize the non-dominated set (NDS) as empty. Then, we execute the loop defined between lines 2–7 for each value $\alpha$. In line 3, we obtain the result from the execution of the model. Then, we get the Makespan and TEC values resulting from the model execution. Then, in line 6, we add the solution obtained to the NDS. Finally, in line 8, the method returns the generated non-dominated set.

---

**Algorithm 1** Weighted sum method.

**input**: $\Delta = \{\alpha_1, \alpha_2, \cdots, \alpha_n\}$, time_limit

**1** NDS ← ∅;

**2 foreach** $\alpha_i \in \Delta$ **do**

**3**   model_result ← `RunWeightedSumModel`($\alpha_i$, time_limit);

**4**   s.Makespan ← `GetMakespan`(model_result);

**5**   s.TEC ← `GetTEC`(model_result);

**6**   NDS ← `AddSolution`(s);

**7 end**

**8 return** NDS ;

---

In the weighted sum method, the decision-maker must define a weight for each objective function. The value of this weight reflects the relative importance of each objective in the overall solution. We adopted several combinations of weights to find the most significant number of optimal Pareto solutions to the problem addressed.

We used the following parameters for Algorithm 1:

• The set $\Delta = \{0, 0.1, 0.2, 0.3, 0.4, 0.5, 0.6, 0.7, 0.8, 0.9, 1\}$ with the possible values for $\alpha$;

• The time limit for each execution of the mathematical model, defined as time_limit $= 800 \times n \times \ln(m)$ sec for each $\alpha$ value, where $m$ is the number of machines and $n$ is the number of jobs;

• $sizeD = 144$: to discretize the day at intervals of 10 min each.

## PROPOSED NSGA-II

The problem addressed belongs to the NP-hard class because it is a generalization of the identical parallel machine scheduling problem, which is NP-hard (*Garey & Johnson, 1979*). For this reason, we used the NSGA-II multi-objective algorithm (*Deb et al., 2002*) to address the problem since there are in literature many reports of successful use of this algorithm (*Deb, UdayaBhaskaraRao & Karthik, 2007*; *Liu et al., 2014*; *Wang et al., 2017*; *Babazadeh et al., 2018*). This algorithm is an alternative to the exact method described in the previous section to find an approximation of the Pareto-optimal front in large instances in an adequate computational time for decision-making.

Algorithm 2 describes how the implemented NSGA-II works.

Algorithm 2 receives the following input parameters: the population size ($size_{pop}$), the probability of mutation ($prob_{mut}$), and the stopping_criterion. In line 2, we created an initial population $P_0$. Then, in the main loop (lines 4–25), we combine the parent $P_t$ and offspring population $Q_t$ to generate a new population $R_t$ (line 5). In line 6, we apply the fast non-dominated sorting method to divide the population $R_t$ into non-dominated sets, called fronts, $\mathcal{F}_1, \mathcal{F}_2, \ldots, \mathcal{F}_k$. A front $\mathcal{F}_i$ dominates another $\mathcal{F}_j$, if and only if, $i < j$ and $R_t = \mathcal{F}_1 \cup \mathcal{F}_2 \ldots \mathcal{F}_k$. In lines 9–13, we select the best frontiers of $\mathcal{F}$ to include in the population $P_{t+1}$. We repeat this procedure as long as it is possible to include a new frontier in $P_{t+1}$ without exceeding the population size. Then we check the size of the population

---

**Algorithm 2** NSGA-II.

**input :**$size_{pop}$, $prob_{mut}$, stopping_criterion

1  $P_0 \leftarrow$ Generate initial population of $size_{pop}$ individuals ;

2  $Q_0 \leftarrow \oslash$;

3  $t \leftarrow 0$ ;

4  **while** stopping_criterion not satisfied **do**

5      $R_t \leftarrow P_t \cup Q_t$ ;

6      $\mathcal{F} \leftarrow$ Fast non-dominated sorting($Rt$ ) ;

7      $P_{t+1} \leftarrow \oslash$ ;

8      $i \leftarrow 1$ ;

9      **while** $|P_{t+1}| + |\mathcal{F}_i| \leq size_{pop}$ **do**

10          Compute *Crowding Distance* of $\mathcal{F}_i$ ;

11          $P_{t+1} \leftarrow P_{t+1} \cup \mathcal{F}_i$;

12          $i \leftarrow i+1$ ;

13      **end**

14      **if** $|P_{t+1}| < size_{pop}$ **then**

15          Sort ($\mathcal{F}_i$, $\prec n$);

16          $j = 1$ ;

17          **while** $|P_{t+1}| < size_{pop}$ **do**

18              $P_{t+1} \leftarrow P_{t+1} \cup \mathcal{F}_i[j]$ ;

19              $j \leftarrow j+1$ ;

20          **end**

21      **end**

22      $Q_{t+1} \leftarrow$ Crossover($P_{t+1}$) ;

23      $Q_{t+1} \leftarrow Q_{t+1} \cup$ Mutation($P_{t+1}$, $prob_{mut}$ ) ;

24      $t \leftarrow t+1$ ;

25  **end**

26  NDS $\leftarrow$ non-dominated solutions of $P_t$ ;

27  **return** NDS ;

---

obtained. If it is not exactly $size_{pop}$, then we order the next frontier $i$ of $\mathcal{F}$ that has not yet been included in $P_{t+1}$, according to the crowding distance, and we select the $size_{pop}$ - $|P_{t+1}|$ first to fill all spaces of the population $P_{t+1}$. In lines 22 and 23, we apply the crossover and mutation operators in $P_{t+1}$ to generate the population $Q_{t+1}$.

The following subsections describe how an initial population is generated and the crossover and mutation operators, respectively.

## Initial population

The initial population of the NSGA-II contains $size_{pop}$ individuals. Two of them are constructed through a greedy strategy, one of which considers only the objective of

**Algorithm 3** Greedy constructive heuristic.

**input:** N,n,m,obj

1 $s \leftarrow \varnothing$;

2 **for** $i = 1$ **to** n **do**

3    $j \leftarrow$ random job $\in$ **N**;

4    **N** $\leftarrow$ **N** $\setminus \{ j\}$;

5    $(i_{best}, \textbf{pos}) \leftarrow$ GreedyChoice(**s**, $j$, **obj**);

6    $s \leftarrow$ Insert(s, $j$, $i_{best}$, **pos**);

7 **end**

8 **return** $s$ ;

minimizing the makespan. In this case, we always choose the operation mode related to the highest speed factor. The other individual considers only the total energy cost. In this case, we choose the operation mode related to the lowest consumption factor. The other individuals ($\textsf{size}_{pop}$-2) of the initial population are randomly generated.

Algorithm 3 describes the greedy strategy for generating an individual to the initial population.

Algorithm 3 starts with an empty initial individual (line 1). The loop between lines 2 and 7 allocates each job $\textsf{n}$ on the machines. Therefore, first, we randomly select a job $j$, which has not yet been allocated (line 3). Then, we identified in the best machine $i_{best}$ of the individual $\textsf{s}$ and the best position $\textsf{pos}$ to insert the job $j$ (line 5). In this case, we consider only one of the objectives of the problem: minimize the makespan or the total energy cost. Then, we allocate job $j$ in position $\textsf{pos}$ of machine $i_{best}$ in individual $\textsf{s}$ (line 6). At the end of this procedure, we return a valid individual $\textsf{s}$ (line 8).

## Crossover

We used the binary tournament selection method to choose each pair of individuals for the crossover operator. We run two tournaments with two individuals each and select the winner of each tournament for the crossover. In our approach, the dominant individual wins the tournament. If both individuals are non-dominated, then we randomly choose an objective and use it to define the winner of the tournament.

Figure 3 illustrates the crossover between two individuals.

After selecting two individuals named parent 1 and parent 2, respectively, we applied the crossover operator to generate new individuals. We adopted the One Point Order Crossover operator from *Vallada & Ruiz (2011)* adapted to the parallel machine problem. We describe its operation below:

1. We define, at random, the crossover points of each machine, as shown in Fig. 3A;

2. We generate two offspring. The first receives the genes to the left of the crossover point defined on each machine of parent 1. The second receives the genes to the right, as shown in Fig. 3B;

3. We mark in parent 2 the genes present in each offspring, as shown in Fig. 3C;

4. We add the unmarked genes of parent 2 to offspring 1 and 2. We add these genes in the position that results in the lowest value for the objective function, whereas this problem has two objective functions, so we randomly select one at each crossover. In the end, we will have two new individuals, as shown in Fig. 3D.

We repeat this procedure until to generate $size_{pop}$ new individuals.

### Mutation

We implemented three mutation operators (Swap, Insert, and Swap of operation mode), described below. These operators maintain the population's genetic diversity and reduce the chances that the algorithm getting stuck at a local optimum.

#### *Swap*

The swap operator works by randomly choosing a job $j_1$, initially allocated in position $a$ on machine $i_1$ and another job $j_2$ allocated in position $b$ of machine $i_2$. Then, we allocate job $j_1$ in position $b$ of machine $i_2$, and we allocate job $j_2$ in position $a$ of machine $i_1$.

Figure 4 illustrates the swap between two jobs $j_1$ and $j_2$. They are initially allocated on machines $i_1$ and $i_2$, respectively. After swapping, we allocate job $j_2$ on machine $i_1$ and job $j_1$ on machine $i_2$.

#### *Insertion*

The insertion operator consists of randomly choosing a job $j_1$ allocated at position $a$ of machine $i_1$ and randomly choosing position $b$ of another machine $i_2$. Then job $j_1$ is removed from machine $i_1$ and inserted into position $b$ on the machine $i_2$.

Figure 5 illustrates this move. The left side shows the scheduling before, and the right side shows it after the insertion move.

#### *Mode change*

In the operation mode change operator, we randomly select a job and change its operation mode at random.

Figure 6 illustrates the application of this operator to the scheduling of offspring 1 of Fig. 3D, which involves 12 jobs. As can be seen, job 8, which is in the sixth position of machine 2, has operation mode 3. After the application of this operator, the job changes to operation mode 1.

The NSGA-II algorithm implemented performs a mutation with a probability equal to $prob_{mut}$.

## COMPUTATIONAL EXPERIMENTS

We coded the NSGA-II algorithm in the C++ language and implemented the mathematical model with the Gurobi 7.0.2 API ( *Gurobi Optimization, 2020*). We performed the tests on a microcomputer with the following configurations: Intel (R) Core (TM) i7-4510U processor with a frequency of 2 GHz, 16 GB of RAM, and 64-bits Ubuntu 19.10 operating system.

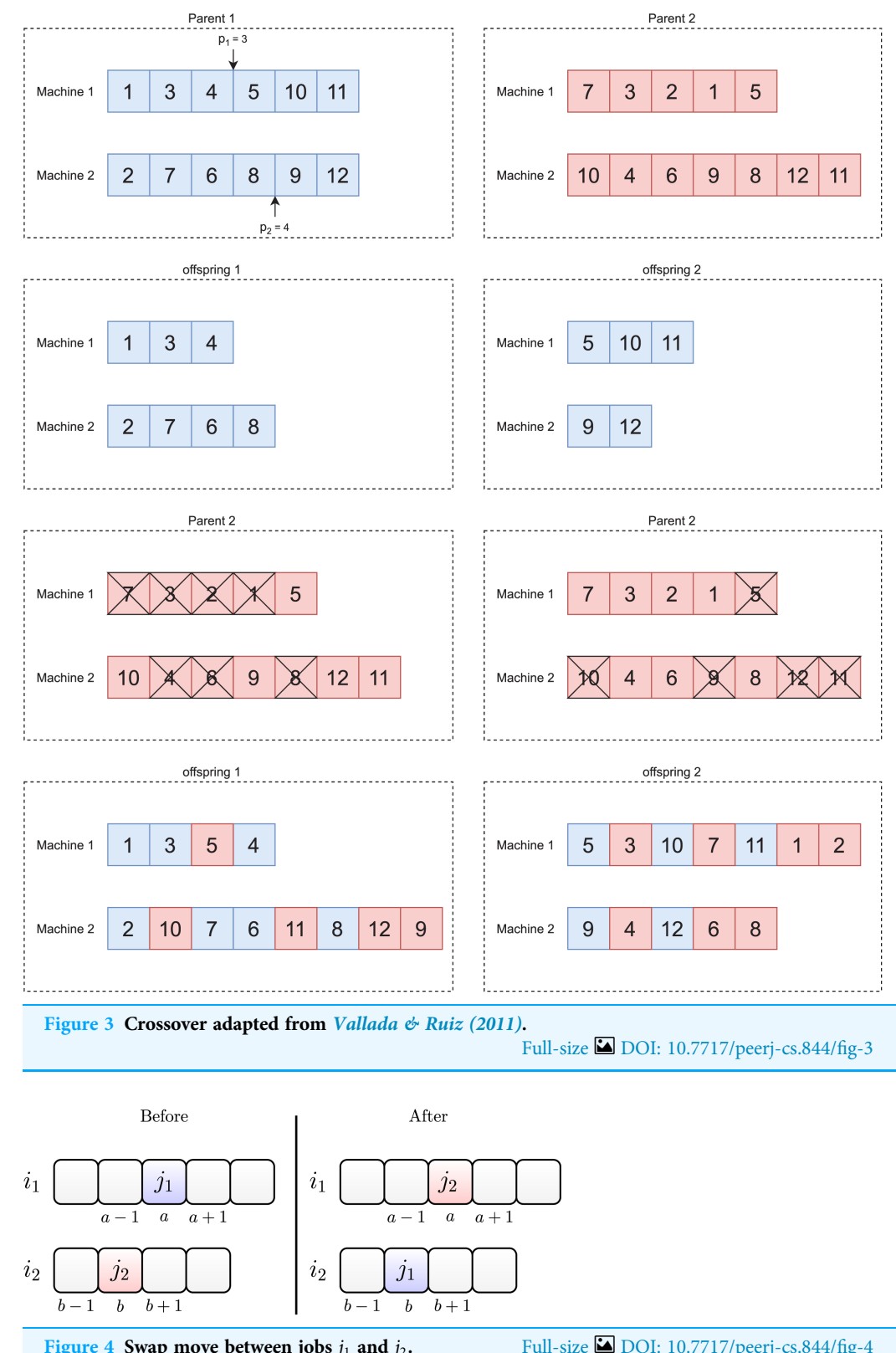

**Figure 3 Crossover adapted from *Vallada & Ruiz (2011)*.**

**Figure 4 Swap move between jobs $j_1$ and $j_2$.**

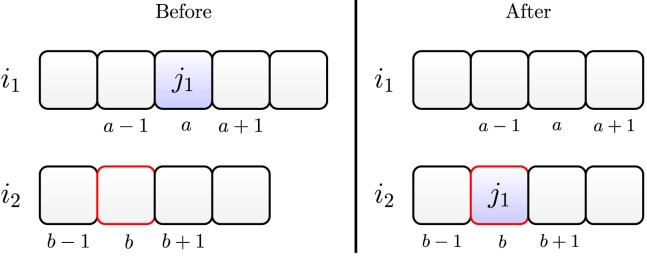

**Figure 5** Insertion move of job $j_1$ on machine $i_2$.

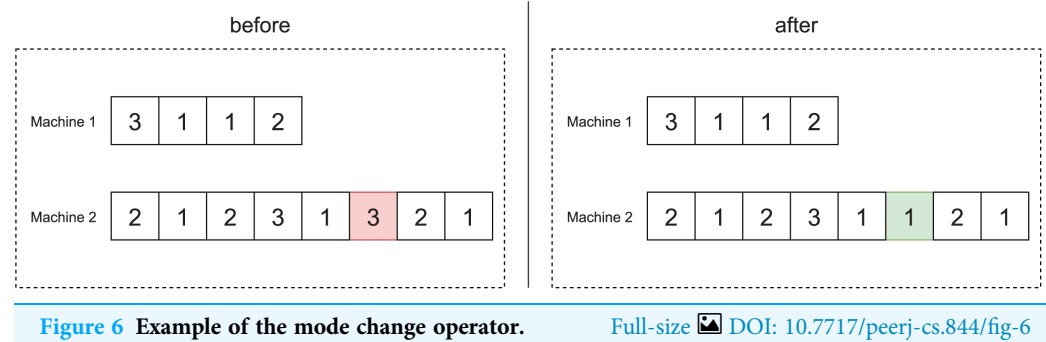

**Figure 6** Example of the mode change operator.

Furthermore, we compared the performance of the NSGA-II algorithm with two basic multi-objective algorithms: MOGA of *Murata & Ishibuchi (1995)* and NSGA-I of *Srinivas & Deb (1994)*. These algorithms use the same NSGA-II crossover and mutation operators described in "Crossover" and "Mutation" and the same stopping criterion.

This section is organized as follows. "Instances Generation" and "Metric description" describe the instances and the metrics used to assess the quality of the set of non-dominated solutions generated by the algorithms. "Tuning of Algorithms' Parameters" shows the parameter calibration of the algorithms. "Results" reports the results.

### Instances generation

Since, as far as we know, there is no set of instances in the literature for the problem addressed, we adapted two instance sets from the literature that deal with similar problems. The first one, called set1, is a subset of the small instances of *Cota et al. (2018)* satisfying the triangular inequality, in which we add information about the energy price on-peak and off-peak hours. The second set, named set2, is also a subset of the large instances of *Cota et al. (2018)*, in which we included instances of 750 jobs. Table 4 shows the characteristics of these sets of instances, which are are available in *Rego, Cota & Souza (2021)*.

### Metric description

The quality of the set of non-dominated solutions found by a method can be analyzed under three aspects: convergence, extension, and distribution. Convergence refers to the proximity of this set to the Pareto-optimal front or to the reference set. In turn, the extension assesses the breadth of the region covered by this set of non-dominated

**Table 4 Instance characteristics.**

| Parameter | set1 | set2 | Based on |
|---|---|---|---|
| $n$ | 6, 7, 8, 9, 10 | 50, 250, 750 | *Vallada & Ruiz (2011)*; *Cota et al. (2018)* |
| $m$ | 2 | 10, 20 | *Vallada & Ruiz (2011)*; *Cota et al. (2018)* |
| $o$ | 3 | 5 | *Mansouri, Aktas & Besikci (2016)*; *Ahilan et al. (2013)*; *Cota et al. (2018)* |
| $P_{ij}$ | $U[1, 99]$ | $U[1, 99]$ | *Vallada & Ruiz (2011)*; *Cota et al. (2018)* |
| $S_{ijk}$ | $U[1, 9]$ | $U[1, 9]$, $U[1, 124]$ | *Vallada & Ruiz (2011)*; *Cota et al. (2018)* |
| $\pi_i$ | $U[40, 200]$ | $U[40, 200]$ | *Cota et al. (2018)* |
| $V_l$ | 1.2, 1, 0.8 | 1.2, 1.1, 1, 0.9, 0.8 | *Mansouri, Aktas & Besikci (2016)*; *Ahilan et al. (2013)*; *Cota et al. (2018)* |
| $\lambda_l$ | 1.5, 1, 0.6 | 1.5, 1.25, 1, 0.8, 0.6 | *Mansouri, Aktas & Besikci (2016)*; *Ahilan et al. (2013)*; *Cota et al. (2018)* |

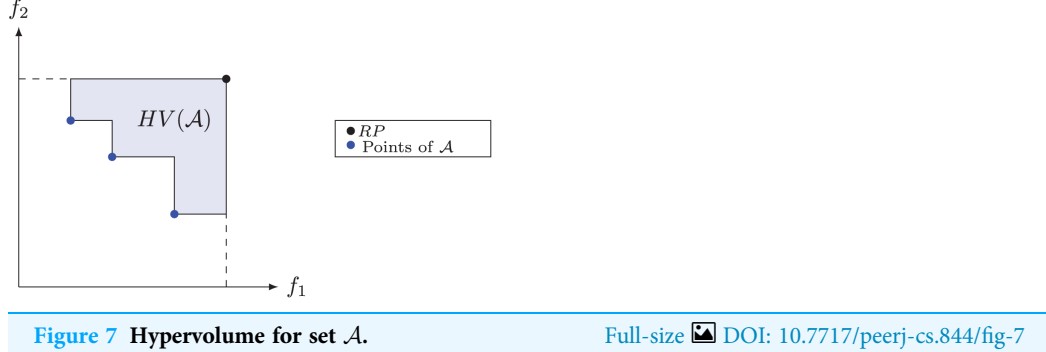

**Figure 7 Hypervolume for set $\mathcal{A}$.**

solutions. Finally, the distribution refers to the uniformity of the spacing between the solutions within the set.

The hypervolume metric is sensitive for convergence and extension and the HCC metric, in turn, is sensitive for distribution and extension.

### Hypervolume

The hypervolume or S metric is a measure of quality often used to compare results from multi-objective algorithms and it was proposed by *Zitzler & Thiele (1998)*. This metric has the ability to provide a combined estimate of convergence and diversity of a set of solutions (*Deb, 2014*). The hypervolume of a non-dominated set measures the area covered or dominated by this set's points, limited by a Reference Point (*RP*). In maximization problems, it is common to use the point (0; 0), while in minimization problems, an upper bound, also known as the Nadir point, is used to limit this area. In Fig. 7, the shaded area defines the hypervolume of the set of non-dominated solutions $\mathcal{A}$ for a problem with two objective functions, in which the point ($max_x$; $max_y$) defines the upper limit. We denote by $HV(\mathcal{A})$ the hypervolume of a set of non-dominated solutions $\mathcal{A}$ relative to a reference point (*Deb, 2014*).

### HCC

Hierarchical cluster counting (HCC) is a metric proposed by *Guimaraes, Wanner & Takahashi (2009)* to evaluate the quality of non-dominated sets that were obtained by

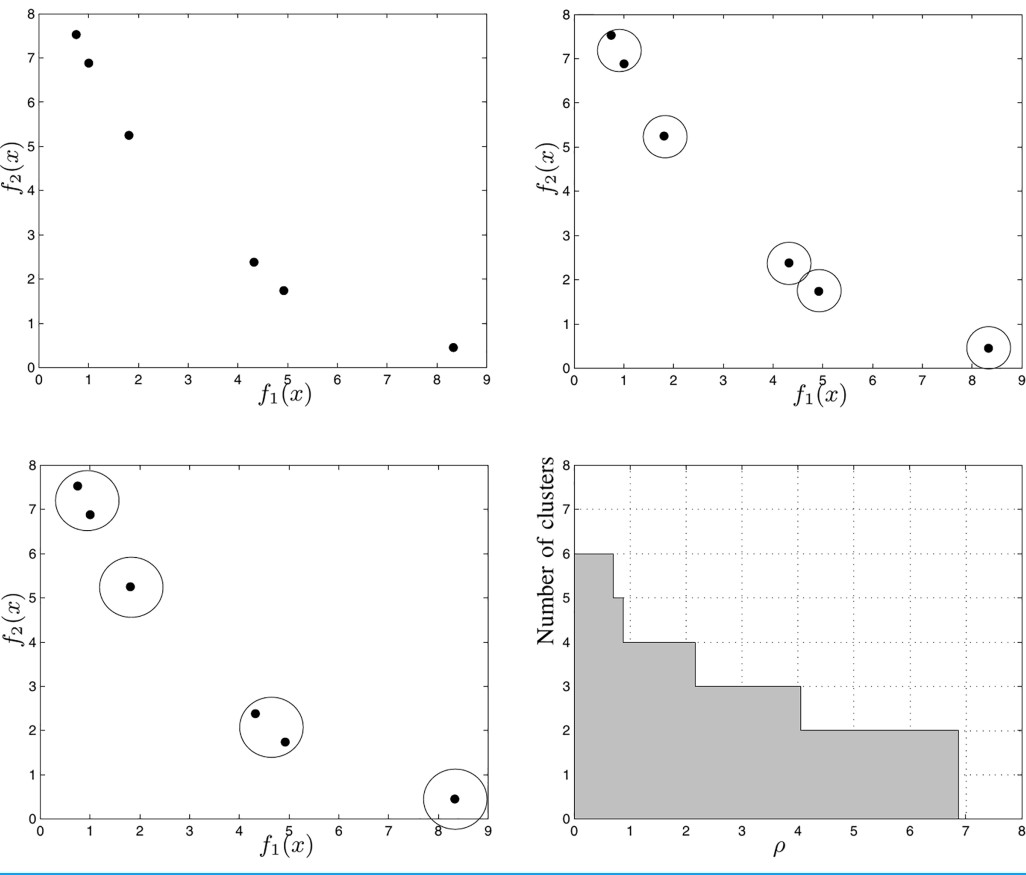

**Figure 8 Example of how to calculate the HCC metric (*Guimaraes, Wanner & Takahashi, 2009*).**

multi-objective optimization algorithms. It is based on hierarchical clustering techniques, such as the Sphere Counting (SC) (*Wanner et al., 2006*) metric. According to *Guimaraes, Wanner & Takahashi (2009)*, the extent and uniformity of a non-dominated set is directly proportional to the HCC value calculated for it.

We calculate the HCC for a set of points $\mathcal{A}$ as follows:

1. Initially, we create a grouping for each point in the set, and consider that each group created is a sphere of radius equal to zero;

2. Then, we calculate the minimum distances of fusion, which is a new assumed value for the radius of the spheres capable of decreasing the number of clusters;

3. We group the points into the same cluster;

4. We repeat steps 2 and 3 until all the points belong to the same grouping;

5. We obtain the HCC value by adding, in each iteration, the product between the distances of fusion and the amount of grouping formed.

Consider Fig. 8, which illustrates the steps to calculate the HCC for a six-point non-dominated set. Figure 8A shows the first cluster in which each point is in a different sphere with radius zero. Figure 8B shows the points grouped into five spheres, each with radius $r_1$.

**Table 5  Test scenarios for algorithms' parameters.**

| Method | Description | Tested values | Irace best value |
|---|---|---|---|
| NSGA-II | Population size ($size_{pop}$) | 80, 90, 100, 110 | 110 |
| | Probability of mutation | 0.04, 0.05, 0.06, 0.07 | 0.05 |
| MOGA | Population size ($size_{pop}$) | 80, 90, 100, 110 | 80 |
| | Probability of mutation | 0.04, 0.05, 0.06, 0.07 | 0.06 |
| NSGA-I | Population size ($size_{pop}$) | 80, 90, 100, 110 | 80 |
| | Probability of mutation | 0.04, 0.05, 0.06, 0.07 | 0.06 |

Figure 8C shows the points grouped into four spheres, each with radius $r_2$. Figure 8D shows, in the Cartesian plane, the relationship between the number of clusters and the the radius of each cluster. The gray region area represents the value of the HCC metric for the set shown in Fig. 8.

## Tuning of algorithms' parameters

The parameter values used in the NSGA-II, MOGA, and NSGA-I algorithms can affect its performance. Therefore, we use the Irace package (*López-Ibáñez et al., 2016*) to find the best values for these parameters. Irace is a software encoded in the R language that automatically performs an iterative procedure to find the most appropriate optimization algorithm settings.

Table 5 shows the test scenarios used. In the first column, we present the description of each NSGA-II parameter; in the second column, the set of values tested for each parameter, and in the third column, the best value returned by Irace.

## Results

In this section, we presented the results of two experiments used to evaluate the NSGA-II algorithm's performance. First, we compare the NSGA-II results with those of the weighted sum method in instances with up to 10 jobs and two machines. Then, we compared the performance of the NSGA-II algorithm with that of the MOGA and NSGA-I algorithms in larger instances, with up to 750 jobs and 20 machines. In both cases, we executed the algorithms 30 times in each instance.

We used the Relative Percentage Deviation ($RPD_i^{HV}$) to evaluate the HV metric for each method *Alg* and instance *i*. It is calculated by Eq. (16):

$$RPD_i^{HV}(Alg) = \frac{HV_i^{RS} - HV_i^{v}}{HV_i^{RS}}, \qquad (16)$$

where $HV_i^{RS}$ is the hypervolume value of the reference set in 30 executions of the algorithm *Alg* in the instance *i*. *v* can assume three values: min, max and *avg*, representing, respectively, the smallest, the largest, and the average of the hypervolume in 30 executions of the algorithm in the instance *i*.

**Table 6  Reference set data in the set1.**

| # ID | # Instance | n | m | HV | RP |
|------|-----------|---|---|-----|-----|
| 1 | 6_2_1439_3_S_1-9 | 6 | 2 | 6,406.67 | (250; 239.91) |
| 2 | 7_2_1439_3_S_1-9 | 7 | 2 | 15,918.87 | (400; 260.68) |
| 3 | 8_2_1439_3_S_1-9 | 8 | 2 | 3,338.29 | (260; 302.58) |
| 4 | 9_2_1439_3_S_1-9 | 9 | 2 | 22,256.33 | (440; 357.69) |
| 5 | 10_2_1439_3_S_1-9 | 10 | 2 | 31,789.09 | (500; 370.68) |

**Table 7  $RPD^{HV}$ and runtime of the methods in the set1.**

| # ID | NSGA-II | | | | | Gurobi | |
|------|---------|---|---|---|---|--------|---|
| | min (%) | max (%) | avg (%) | sd | time (s) | UB (%) | time (s) |
| 1 | 0.00 | 0.00 | 0.00 | 0.00 | 4.16 | 1.89 | 172.09 |
| 2 | 0.68 | 0.68 | 0.68 | 0.00 | 4.85 | 1.12 | 549.86 |
| 3 | 0.00 | 0.00 | 0.00 | 0.00 | 5.54 | 0.34 | 2,140.40 |
| 4 | 0.00 | 0.75 | 0.62 | 0.28 | 6.24 | 5.12 | 8,312.92 |
| 5 | 0.66 | 0.89 | 0.66 | 0.04 | 6.93 | 1.51 | 39,396.63 |

**Table 8  HCC and runtime of the methods in the set1.**

| # ID | NSGA-II | | | | | Gurobi | |
|------|---------|---|---|---|---|--------|---|
| | min | max | avg | sd | time (s) | UB | time (s) |
| 1 | 305.16 | 305.16 | 305.16 | 0.00 | 4.16 | 253.78 | 172.09 |
| 2 | 75.87 | 75.87 | 75.87 | 0.00 | 4.85 | 43.52 | 549.86 |
| 3 | 356.33 | 356.33 | 356.33 | 0.00 | 5.54 | 215.71 | 2,140.40 |
| 4 | 70.63 | 80.64 | 78.97 | 12.80 | 6.24 | 6.56 | 8,312.92 |
| 5 | 84.98 | 87.79 | 87.70 | 2.05 | 6.93 | 77.15 | 39,393.63 |

### NSGA-II × Gurobi

In this section, we reported the results of the NSGA-II algorithm and the exact method in the set of instances set1.

Table 6 shows the reference set data for these instances. In this table, the first two columns show the instance identifier and name, respectively. The next two columns show the number of jobs and machines, respectively. The fifth column presents the hypervolume of this reference set. Finally, the last column presents the reference point ($C_{max}$; *TEC*) used to calculate the hypervolume of each instance.

Tables 7 and 8 show the method results concerning the $RPD^{HV}$ and the HCC metrics. In these tables, the first column identifies the instance. The next three columns show the minimum, maximum and average values of the $RPD^{HV}$ and HCC, respectively, concerning the NSGA-II method. The fifth column shows the standard deviation of the results. The seventh column shows the upper bound (UB) returned by the exact method

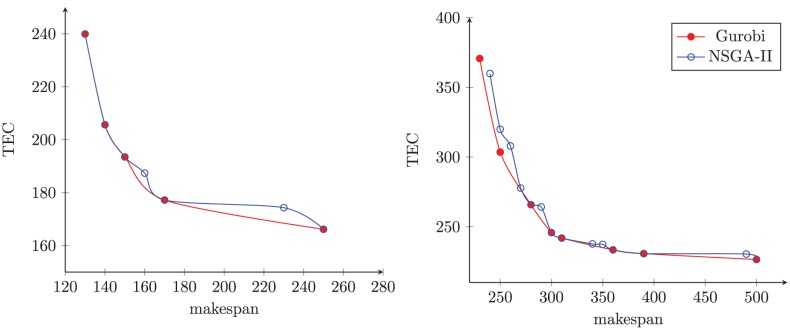

**Figure 9  Frontiers found by NSGA-II and Gurobi methods.**

concerning the $RPD^{HV}$ and HCC metrics. Finally, the sixth and eighth columns show the times, in seconds, of the NSGA-II and the exact method, respectively.

We can see in Table 6 that in the set of instances set1, the $RPD^{HV}$ of the NSGA-II algorithm is lower in all comparisons (min, max e *avg*) compared to the exact method. We can also verify that the standard deviation of the NSGA-II algorithm in instances with up to 8 jobs (ID 1, 2, 3) is equal to zero. In other words, in these instances, all NSGA-II executions obtained the same set non-dominated. We also can note that the execution time of the NSGA-II is much less than that of the exact algorithm.

Concerning Table 8, we noted that the NSGA-II algorithm has a higher HCC value than the exact method in all comparisons. Thus, we can conclude that the non-dominated set of the NSGA-II method has better diversity and uniformity.

Figure 9 presents the non-dominated sets obtained by an NSGA-II execution and the other using the exact method in two randomly selected instances. The first instance has six jobs and two machines, and the second has 10 jobs and two machines. In this figure, the blue dots represent the solutions of the NSGA-II, and the red dots represent the solutions of the exact method. The *x* axis represents the makespan, and the *y* axis represents the total energy cost.

We can notice in Fig. 9A that the NSGA-II non-dominated set contains all the solutions found by the exact method, plus two additional solutions. In this example, the two methods have the same amplitude, and the NSGA-II was able to find a set of solutions with higher cardinality. On the other hand, Fig. 9B shows that the non-dominated set contains six of the eight solutions found by the exact method and eight other solutions. In this example, the exact method showed better amplitude than the NSGA-II, but this obtained higher cardinality than the exact method.

Considering these results, we observed that the NSGA-II finds good quality solutions and requires less computational time than the exact method.

### NSGA-II in large instances compared with other literature algorithms

Here, we presented the results of the NSGA-II, MOGA, and NSGA-I algorithms in the set of instances set2.

Table 9 shows the reference set data for the instances of set2. Its organization follows the same description as the previous section's tables.

**Table 9 Reference set data in the set2.**

| # ID | # Instance | $n$ | $m$ | HV | RP |
|------|-----------|-----|-----|-----|-----|
| 1 | 50_10_1439_5_S_1-9 | 50 | 10 | 56,872 | (280; 452.653) |
| 2 | 50_10_1439_5_S_1-124 | 50 | 10 | 181,673 | (456; 909.559) |
| 3 | 50_20_1439_5_S_1-9 | 50 | 20 | 15,933 | (114; 323.339) |
| 4 | 50_20_1439_5_S_1-124 | 50 | 20 | 151,884 | (392; 642.568) |
| 5 | 250_10_1439_5_S_1-9 | 250 | 10 | 1,488,058 | (1,457; 2,245.16) |
| 6 | 250_10_1439_5_S_1-124 | 250 | 10 | 6,986,225 | (4,374; 2,930.02) |
| 7 | 250_20_1439_5_S_1-9 | 250 | 20 | 526,773 | (420; 2,570.86) |
| 8 | 250_20_1439_5_S_1-124 | 250 | 20 | 1,383,849 | (988; 3,179.8) |
| 9 | 750_10_1439_5_S_1-9 | 750 | 10 | 7,678,168 | (3,665; 5,630.8) |
| 10 | 750_10_1439_5_S_1-124 | 750 | 10 | 105,897,123 | (19,400; 9,442.33) |
| 11 | 750_20_1439_5_S_1-9 | 750 | 20 | 3,791,005 | (1,364; 5,573.7) |
| 12 | 750_20_1439_5_S_1-124 | 750 | 20 | 34,971,819 | (7,740; 7,065.71) |

**Table 10 $RPD^{HV}$ and runtime of the algorithms in the set2.**

| # ID | MOGA | | | | NSGA-I | | | | NSGA-II | | | | |
|------|---------|---------|---------|------|---------|---------|---------|------|---------|---------|---------|------|----------|
| | min (%) | max (%) | avg (%) | sd | min (%) | max (%) | avg (%) | sd | min (%) | max (%) | avg (%) | sd | time (s) |
| 1 | 0.08 | 0.28 | 0.19 | 0.06 | 0.04 | 0.18 | **0.10** | 0.03 | 0.01 | 0.18 | **0.10** | 0.03 | 115.13 |
| 2 | 0.03 | 0.20 | 0.12 | 0.04 | 0.05 | 0.20 | 0.11 | 0.04 | 0.03 | 0.20 | **0.10** | 0.04 | 115.13 |
| 3 | 0.04 | 0.27 | 0.15 | 0.06 | 0.07 | 0.25 | 0.15 | 0.05 | 0.05 | 0.21 | **0.12** | 0.05 | 149.79 |
| 4 | 0.04 | 0.16 | 0.11 | 0.03 | 0.02 | 0.16 | 0.10 | 0.03 | 0.02 | 0.17 | **0.09** | 0.03 | 149.79 |
| 5 | 0.22 | 0.27 | 0.24 | 0.02 | 0.02 | 0.09 | 0.06 | 0.02 | 0.01 | 0.04 | **0.03** | 0.01 | 575.65 |
| 6 | 0.01 | 0.11 | 0.06 | 0.03 | 0.01 | 0.04 | **0.02** | 0.01 | 0.01 | 0.03 | **0.02** | 0.01 | 575.65 |
| 7 | 0.13 | 0.36 | 0.24 | 0.06 | 0.05 | 0.14 | 0.10 | 0.02 | 0.01 | 0.08 | **0.05** | 0.02 | 748.93 |
| 8 | 0.03 | 0.11 | **0.07** | 0.02 | 0.04 | 0.12 | 0.08 | 0.02 | 0.04 | 0.12 | **0.07** | 0.02 | 748.93 |
| 9 | 0.21 | 0.33 | 0.24 | 0.03 | 0.02 | 0.06 | 0.04 | 0.01 | 0.01 | 0.02 | **0.01** | 0.00 | 1,726.94 |
| 10 | 0.10 | 0.15 | 0.13 | 0.01 | 0.01 | 0.03 | 0.02 | 0.00 | 0.01 | 0.02 | **0.01** | 0.00 | 1,727.94 |
| 11 | 0.13 | 0.21 | 0.17 | 0.02 | 0.05 | 0.17 | 0.10 | 0.03 | 0.01 | 0.05 | **0.02** | 0.01 | 2,246.80 |
| 12 | 0.04 | 0.07 | 0.06 | 0.01 | 0.01 | 0.02 | 0.02 | 0.00 | 0.00 | 0.02 | **0.01** | 0.00 | 2,246.80 |

**Note:**
The best average values are highlighted in bold.

Tables 10 and 11 report the $RPD^{HV}$ and HCC values, respectively, of the algorithms in the set of instances set2. As can be seen, NSGA-II achieved the best average results regarding hypervolume in all instances. On the other hand, it won in 2/3 of the instances concerning the HCC. These results indicate that the NSGA-II algorithm outperforms MOGA and NSGA-I algorithms concerning these metrics.

Figures 10A and 10B illustrate the Pareto front obtained from each algorithm in two different instances. The first instance has 50 jobs and 20 machines, and the second has 750 jobs and 10 machines. As can be seen, the NSGA-II produced sets of non-dominated

**Table 11 HCC and runtime of the algorithms in the set2.**

| # ID | MOGA | | | | NSGA-I | | | | NSGA-II | | | | Time (s) |
|------|------|-----|-----|-----|--------|-----|-----|-----|---------|-----|-----|-----|----------|
| | min | max | *avg* | *sd* | min | max | *avg* | *sd* | min | max | *avg* | *sd* | |
| 1 | 555.75 | 835.13 | 701.86 | 89.97 | 1,181.10 | 1,654.00 | 1,422.89 | 129.12 | 1,253.90 | 1,715.60 | **1,446.94** | 128.44 | 115.13 |
| 2 | 846.86 | 2,116.50 | 1,503.49 | 278.61 | 1,234.80 | 2,466.60 | **1,844.78** | 343.75 | 1,063.80 | 2,224.00 | 1,653.94 | 323.72 | 115.13 |
| 3 | 171.46 | 639.33 | 401.74 | 97.33 | 210.11 | 2,466.60 | 397.67 | 106.53 | 279.32 | 659.35 | **431.34** | 89.45 | 149.79 |
| 4 | 1,343.70 | 2,357.30 | 1,838.98 | 271.90 | 1,251.30 | 573.29 | **1,885.60** | 240.14 | 1,325.60 | 2,530.40 | 1,861.56 | 276.50 | 149.79 |
| 5 | 713.06 | 1,217.10 | 937.12 | 105.39 | 6,684.20 | 573.29 | 8,270.10 | 788.31 | 8,268.30 | 10,702.62 | **9,187.19** | 535.43 | 575.65 |
| 6 | 1,483.80 | 10,998.41 | 5,175.87 | 2,765.31 | 9,301.70 | 2,402.80 | 10,844.27 | 1,222.07 | 9,866.60 | 12,470.58 | **11,042.90** | 669.22 | 575.65 |
| 7 | 448.98 | 1,231.10 | 762.16 | 226.50 | 1,726.60 | 2,402.80 | **3,429.42** | 903.29 | 1,878.40 | 4,193.80 | 2,556.17 | 514.60 | 748.93 |
| 8 | 6,488.40 | 9,807.20 | 8,451.12 | 954.44 | 6,942.00 | 9,516.80 | 8,739.29 | 864.36 | 7,360.40 | 10,702.12 | **8,790.46** | 872.03 | 748.93 |
| 9 | 1,441.00 | 2,348.90 | 1,796.00 | 220.21 | 17,117.09 | 9,516.80 | 19,413.19 | 946.57 | 20,218.49 | 23,418.80 | **21,967.44** | 858.86 | 1,726.94 |
| 10 | 9,766.60 | 26,604.74 | 18,781.83 | 3,936.60 | 50,872.40 | 14,703.88 | **61,504.40** | 9,730.96 | 53,824.74 | 75,091.86 | 60,472.87 | 6,184.92 | 1,726.94 |
| 11 | 900.31 | 1,775.60 | 1,386.85 | 199.20 | 8,076.50 | 14,703.88 | 11,851.43 | 1,450.16 | 10,215.91 | 15,284.62 | **12,925.00** | 1,473.66 | 2,246.80 |
| 12 | 6,576.30 | 14,153.94 | 10,179.42 | 2,092.55 | 25,197.32 | 5,275.80 | 29,611.39 | 3,419.31 | 26,060.88 | 33,175.70 | **29,703.90** | 1,663.63 | 2,246.80 |

Note:
The best average values are highlighted in bold.

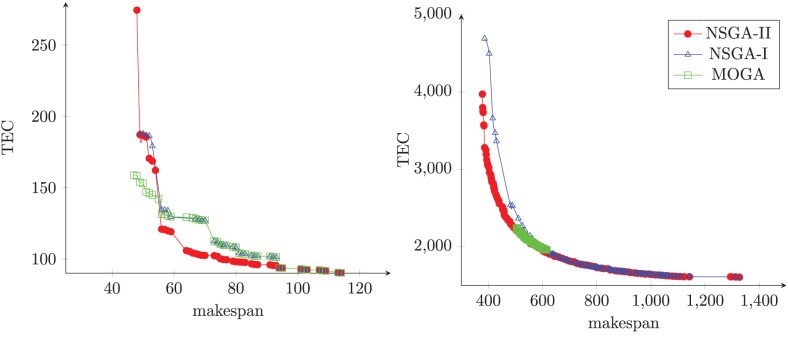

**Figure 10 The Pareto front obtained from each algorithm.**

solutions with good convergence, diversity, uniformity, and amplitude when compared with others algorithms.

Figures 11 and 12 show the boxplot of the RPD and HCC results, respectively.

To verify if the differences between the results presented by the algorithms are statistically significant, we performed the hypothesis tests below.

$$\begin{cases} H_0 : \mu_1 = \mu_2 = \mu_3 \\ H_1 : \exists i,j \mid \mu_i \neq \mu_j \end{cases}$$

In the first test, $\mu_1$, $\mu_2$, and $\mu_3$ represent the average $RPD^{HV}$ for NSGA-II, MOGA, and NSGA-I, respectively. In the second test, $\mu_1$, $\mu_2$, and $\mu_3$ represent the average HCC for the algorithms in the same sequence.

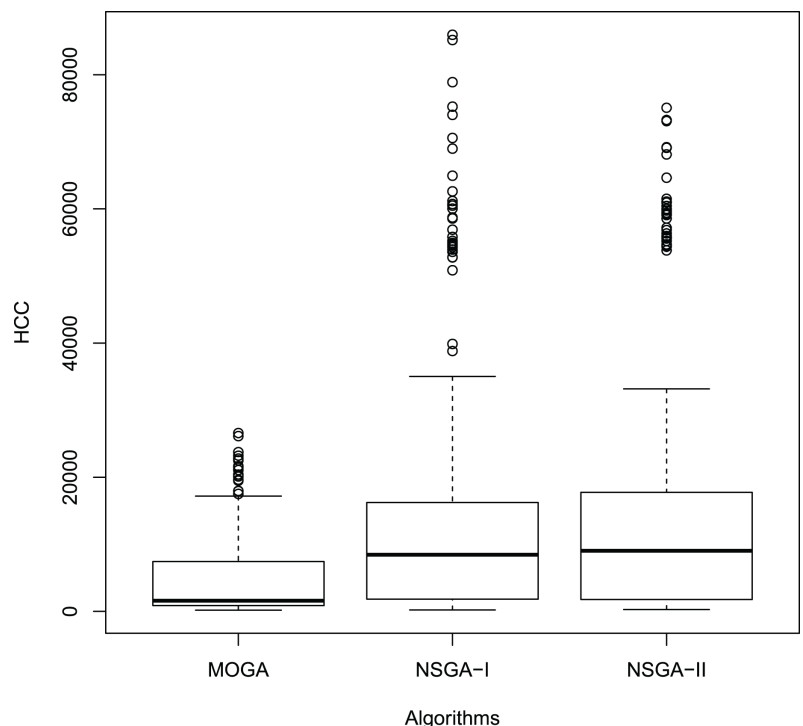

**Figure 11 Boxplot of the $RPD^{HV}$ results.** 

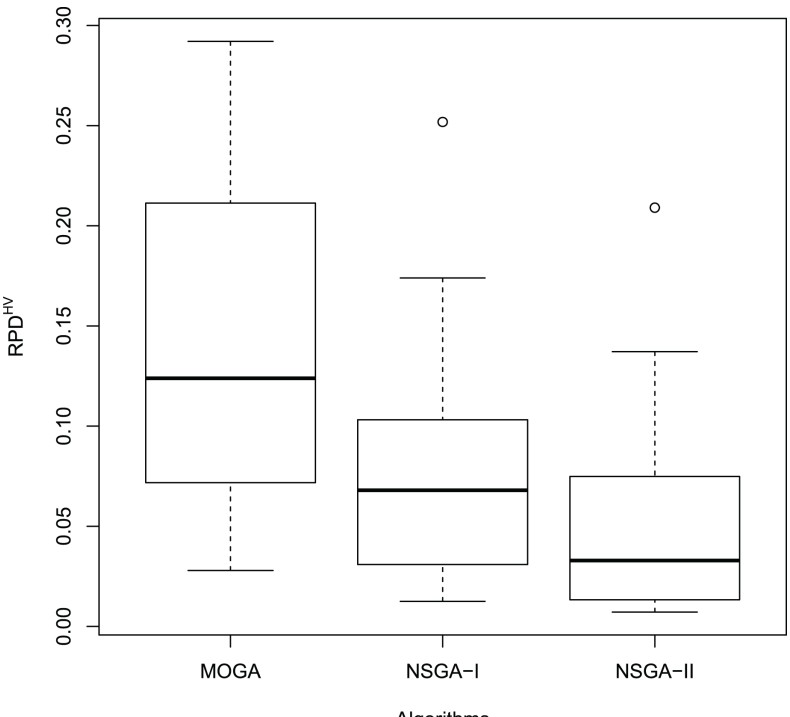

**Figure 12 Boxplot of the HCC results.** 

**Table 12 p-values of the Shapiro-Wilk normality test concerning $RPD^{HV}$ and HCC values.**

| Algorithm | $RPD^{HV}$ | HCC |
|---|---|---|
| MOGA | 0.0072 | 2.2e−16 |
| NSGA-I | 0.0002 | 2.2e−16 |
| NSGA-II | 2.289e−06 | 2.2e−16 |

**Table 13 p-values of the paired Wilcoxon signed-rank test concerning $RPD^{HV}$ and HCC values ($\alpha = 0.05$).**

| Comparison | $RPD^{HV}$ | HCC |
|---|---|---|
| MOGA *vs* NSGA-I | 8.0e−08 | 2e−16 |
| MOGA *vs* NSGA-II | 7.8e−10 | 2e−16 |
| NSGA-I *vs* NSGA-II | 1.8e−08 | 0.0001 |

Before performing the hypothesis tests, we need to choose the test type, parametric or non-parametric. Generally, parametric tests are more powerful; however, to use them, it is necessary to satisfy three assumptions:

1. Normality: every sample must originate from a population with normal distribution,

2. Independence: the samples must be independent of each other,

3. Homoscedasticity: there must be equality of variances across samples.

We applied the Shapiro-Wilk normality test to the samples with the $RPD^{HV}$ and HCC values from each algorithm and showed its results in Table 12.

With confidence level of 95% ($\alpha = 0.05$), we can say that the results presented in Table 12 not present evidence that the results of the algorithms come from a population with normal distribution.

Thus, we applied the paired Wilcoxon signed-rank non-parametric test (*Wilcoxon, 1945*). Table 13 reports the results of this test obtained by the NSGA-II, MOGA, and NSGA-I algorithms for the samples of the $RPD^{HV}$ and HCC values.

According to Table 13, there are significant statistical difference between each pair of algorithms. Thus, these tests confirm the results in Tables 10 and 11, indicating that NSGA-II outperforms both MOGA and NSGA-I.

## CONCLUSIONS

This paper addressed the unrelated parallel machine scheduling problem with sequence-dependent setup times for minimizing the total energy cost and the makespan.

To solve it, we developed a mixed-integer linear programming formulation and applied the weighted sum method to generate sets of non-dominated solutions to the problem. Considering that this formulation could not solve larger instances of the problem, we adapted the NSGA-II algorithm to deal with them.

To test the two solution methods, we adapted instances of the literature to contemplate all the problem's characteristics. We divided these instances into two groups. The first

group consists of small instances with up to 10 jobs and 2 machines, while the second group contains large instances, with up to 750 jobs and 20 machines. We evaluated the methods concerning the hypervolume and HCC metrics.

Initially, we used part of the set of instances to tuning the parameter values of the NSGA-II algorithm. To this end, we used the Irace package.

We validated the NSGA-II algorithm in small instances, comparing its results with those produced by the exact method. The NSGA-II algorithm showed good convergence and diversity. Besides, it spent much shorter CPU time than that required by the exact method.

In large instances, the results showed that the NSGA-II outperforms, with 95% confidence level, MOGA and NSGA-I algorithms concerning the hypervolume and HCC metrics. Thus, the proposed algorithm finds non-dominated solutions with good convergence, diversity, uniformity, and amplitude.

As future work, we suggest testing other crossover and mutation operators for the NSGA-II. Besides, we intend to implement other multi-objective algorithms, such as Strength Pareto Evolutionary Algorithm 2 (SPEA2), Niched Pareto Genetic Algorithm (NPGA), Pareto Envelope-based Selection Algorithm II (PESA-II), Multi-objective Variable Neighborhood Search (MOVNS), and Multi-objective Evolutionary Algorithm Based on Decomposition (MOEA/D).

## ACKNOWLEDGEMENTS

The authors thank the Universidade Federal de Ouro Preto, the Universidade Federal dos Vales do Jequitinhonha, and the Instituto Tecnológico Vale for the use of the physical infrastructure and equipment used to produce this work. They also thank the Universidade Federal dos Vales do Jequitinhonha for granting Marcelo Ferreira Rego the temporary removal of his role as professor of the institution.

### Funding

This work was supported by the Coordenação de Aperfeiçoamento de Pessoal de Nível Superior-Brazil (CAPES)-Finance Code 001, Fundação de Amparo à Pesquisa do Estado de Minas Gerais (FAPEMIG, grant 676-17), and Conselho Nacional de Desenvolvimento Científico e Tecnológico (CNPq, grant 303266/2019-8). The funders had no role in study design, data collection and analysis, decision to publish, or preparation of the manuscript.

### Grant Disclosures

The following grant information was disclosed by the authors:
Coordenação de Aperfeiçoamento de Pessoal de Nível Superior- Brazil: 001.
Fundação de Amparo à Pesquisa do Estado de Minas Gerais: 676-17.
Conselho Nacional de Desenvolvimento Científico e Tecnológico: 303266/2019-8.

## Competing Interests

The authors declare that they have no competing interests.

## Author Contributions

- Marcelo F. Rego conceived and designed the experiments, performed the experiments, analyzed the data, performed the computation work, prepared figures and/or tables, authored or reviewed drafts of the paper, and approved the final draft.
- Júlio Cesar E. M. Pinto conceived and designed the experiments, performed the experiments, analyzed the data, performed the computation work, prepared figures and/or tables, authored or reviewed drafts of the paper, and approved the final draft.
- Luciano P. Cota conceived and designed the experiments, performed the experiments, analyzed the data, performed the computation work, prepared figures and/or tables, authored or reviewed drafts of the paper, and approved the final draft.
- Marcone J. F. Souza conceived and designed the experiments, performed the experiments, analyzed the data, performed the computation work, prepared figures and/or tables, authored or reviewed drafts of the paper, and approved the final draft.

## Data Availability

Data is available at GitHub: https://github.com/marcelofr/UPMSP_ME_INSTANCE.

## Supplemental Information

Supplemental information for this article can be found online at http://dx.doi.org/10.7717/peerj-cs.844#supplemental-information.

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
