# Peer review of "A mathematical formulation and an NSGA-II algorithm for minimizing the makespan and energy cost under time-of-use electricity price in an unrelated parallel machine scheduling"

_PeerJ Computer Science, doi:10.7717/peerj-cs.844_

## Round 0.1 · original submission · Major Revisions

It is recommended to discuss the basic issues in detail and to compare the proposed method with a basic algorithm to measure its quality.

The authors state that they modified the NSGA-II and applied it to a real-world problem. However, the performance of the modified algorithm should be tested by comparing it with some of the basic multiobjective algorithms such as MOGA, MOMGA, NPGA, NSGA, NSGA-II, PAES, PESA, PESA-II, SPEA, SPEA2.

·

Basic reporting

The language used in the study is very clear and unambigous. In the introduction, the topics are discussed regularly. There are enough figures to make understandable. Especially the part experiment and test has been prepared in detail and with care. Necessary definitions that make the equations more understandable are given in tables.

Experimental design

It is a study in accordance within Aims and scope of the journal.

Validity of the findings

The conclusion part is discussed in detail and explanatory. The curves obtained prove the effectiveness of the proposed method.

Reviewer 2 ·

Basic reporting

no comment

Experimental design

no comment

Validity of the findings

no comment

Additional comments

In this study, which optimizes the scheduling problem of a new bi-objective unrelated parallel machines with sequential setup times, it is aimed to minimize the makespan and the total energy cost.

The motivation, importance and flow of the article is quite good. The contribution of the article appears to have met the standards for the "PeerJ Computer Science" journal.

Reviewer 3 ·

Basic reporting

The abstract should not be expressed in general sentences. Instead, it should be explained as much as possible about the work done and the conditions under which the work takes place. Effects, values, results, etc. it will be much more convenient and useful for the reader/researcher to provide information containing answers to questions.

The introduction section should be developed more clearly and concretely for those who are going to read this article. There are decouples between paragraphs. It should be rewritten more fluently.

Experimental design

Similar articles in this field should be summarized and tabulated for readers at the end of the literature review section. A flowchart should be attached to the article that shows the general structure of the work. The sections proposed in the flowchart can be shown in more detail.

The dataset studied and given as an example should be specified in the article and added as a reference. The success of the NSGA-II algorithm should be compared with other basic algorithms in this field. The advantages and disadvantages should be revealed by examining the results obtained.

Validity of the findings

It should be compared with similar results obtained in similar studies conducted earlier. The effective aspects of the study should be highlighted. Considering the situations arising as a result of this study, new ideas should be offered to readers.

Additional comments

no comment

---

## Round 0.2 · accepted · Accept

The authors made the requested corrections and fulfilled their obligations.

Reviewer 3 ·

Basic reporting

The authors correctly understood the changes I recommended and responded appropriately to all of them. Considering the necessary explanations, the article was revised as desired.

Experimental design

The authors correctly understood the changes I recommended and responded appropriately to all of them. Considering the necessary explanations, the article was revised as desired.

Validity of the findings

The authors correctly understood the changes I recommended and responded appropriately to all of them. Considering the necessary explanations, the article was revised as desired.

Additional comments

The authors correctly understood the changes I recommended and responded appropriately to all of them. Considering the necessary explanations, the article was revised as desired.